# Friendships in Integrative Settings: Network Analyses in Organized Sports and a Comparison with School

**DOI:** 10.3390/ijerph18126603

**Published:** 2021-06-19

**Authors:** Alexander Steiger, Fabian Mumenthaler, Siegfried Nagel

**Affiliations:** 1Institute of Sport Science, University of Bern, 3012 Bern, Switzerland; fabian.mumenthaler@phbern.ch (F.M.); siegfried.nagel@ispw.unibe.ch (S.N.); 2Institute of Special Education, Bern University of Teacher Education, 3012 Bern, Switzerland

**Keywords:** social network analysis, psychosocial health, social participation, physical activity, sports, children, intellectual disability, inclusion, exponential random graph models

## Abstract

Social networks affect health. In this empirical study, friendship networks in integrative organized sports were examined and then compared with friendship networks in integrative school. Relevant factors for friendship network formation were investigated, with a particular interest in the relevance of intellectual disability. Advanced social network analysis was performed using exponential random graph modeling (ERGM) on individual attributes and dyadic factors, while controlling for network structures. A meta-analysis of estimated ERGMs in each setting, organized sports and school, was conducted. When controlling for all other included factors, intellectual disability is not relevant for friendship networks in organized sports. Athletic ability and gender homophily are relevant factors, while language and similarity in athletic ability are not. Contrary to the results for organized sports, intellectual disability and speaking a foreign language at home are negative factors in friendship networks at school. Athletic ability is important in both settings. Regarding dyadic factors, gender homophily is important in both settings, but similarity in athletic ability is not. To foster the psychosocial health of children with intellectual disabilities, they should be encouraged to participate in integrative organized sports as, there, they are part of friendship networks in a manner equal to their peers without an intellectual disability.

## 1. Introduction

Social networks affect health through different mechanisms, e.g., provision of social support or social influence of peers [1,2,3]. Especially positive relationships, such as friendships, promote positive outcomes in social, cognitive and emotional development [4,5]. However, to experience the benefits of social networks, one has to be part of a group first [6]. People with intellectual disabilities (ID), in particular, experience health inequalities and social exclusion in many societies [7,8,9]. They have been identified as having more restricted social networks than people with physical disabilities and without disabilities [10,11]. The Convention on the Rights of Persons with Disabilities (CRPD) addresses these inequalities and aims to facilitate the social participation of people with disabilities.

Concerning social networks of children with ID, research usually focuses on school. In school, reports on children with ID being at risk of social exclusion are predominant [12,13,14,15,16]. However, the CRPD not only seeks to ensure social participation in school but in all life domains. Despite this, research efforts in other life domains, such as leisure and sports (CRPD, article 30), remain limited.

Organized sports (an umbrella term for extracurricular sports activities provided by voluntary sports clubs, for-profit organizations, schools and municipalities) can help people with disabilities to form social networks, which contribute to social participation in and beyond sports [17,18]. Although physical health can be fostered through any form of sports, organized sports mainly and strongly contribute to psychosocial health [19]. Especially when providing sporting activities in an integrative setting—i.e., people with and without disabilities practicing sports together—organized sports are often mentioned as enablers of social participation for people with disabilities [20,21,22,23]. However, concerns are also voiced about whether and how organized sports can effectively deliver what is expected of them [24]. Much remains unknown about social networks in integrative organized sports [25,26,27]. Studies on the friendships of children with ID in organized sports mainly consist of Unified Sports program evaluations, and report ambivalent findings [25,26]. Although some report successful friendships between children with and without ID [28,29,30,31,32], other studies indicate a lack of sustainable contacts [33,34]. Quantitative large-scale studies and research other than program evaluations seem to be nonexistent [26,35].

Simply placing children with and without ID in the same group does not automatically result in friendships beyond (dis)ability in either organized sports or school. This raises questions as to how friendship networks are formed in integrative settings: Which factors are relevant for friendship networks? Are relevant factors different in integrative organized sports and school? What role does ID play in friendship networks?

### 1.1. Network Structures: Theoretical Framework

Friendship networks are usually investigated using social network analysis (SNA). However, how SNA has been conducted in integrative settings, with few exceptions (e.g., [12]), neglects the theoretical assumptions of social networks. First, when investigating social network data, interdependencies must be considered [36]. In social network research, there is a consensus that the ties between actors are dependent on other ties in the network (endogenous factors) [36,37]. Thus, a friendship tie between two children influences other ties, i.e., it creates, maintains or destroys other ties in the network [38,39]. Two endogenous factors that structure friendship networks are reciprocity and transitivity [38,39,40,41,42]. Reciprocity indicates the prevalence of mutual friendship nominations in a friendship network. Some definitions of friendship even view reciprocity as an essential part thereof [43]. Transitivity denotes the tendency of networks to close triads, i.e., “my friend’s friend is also my friend”. Furthermore, ties between actors are dependent on actor attributes and similarities of attributes (exogenous factors) [36,37]. For instance, a friendship network is, to some extent, structured by children’s individual attributes (e.g., individual abilities) and similarities in attributes (e.g., similar ability level). Second, those multiple social processes operate simultaneously [36]. A friendship network can, for instance, be structured at the same time by popular individuals and by gender homophily. Third, observed social processes can be seen as a proof of ongoing social processes in a network [36,38]. If triadic closure is a significant social process in a network, further triads will likely be formed in the future. Fourth, it is assumed that social networks are locally emergent, i.e., multiple social processes have different prevalences in different parts of the network [36]. Consequently, social network structures emerge and develop locally. Fifth, although social networks are structured by endogenous and exogenous factors, new ties can still develop randomly [36].

The described theoretical assumptions have implications on SNA. These assumptions are violated using traditional procedures like regression analyses. Contrarily, advanced SNA (e.g., exponential random graph [36] or stochastic actor-oriented models [44]) comply with these theoretical assumptions [36].

### 1.2. Network Structures: Empirical Findings

Empirical studies have identified similarities of attributes as relevant and robust network structures [45,46]. The often-observed tendency of people with similar attributes to cluster, which is colloquially described as “birds of a feather flock together”, is usually referred to as the “homophily principle” [45,47]. A prominent attribute for homophily in friendship networks is gender: girls tend to befriend girls and boys befriend boys. While gender homophily has thoroughly been observed in school [48,49,50,51], it has, to the best of our knowledge, not yet been investigated in organized sports. In a recent network study on the differences between girls’ and boys’ sports groups, approximately one-third were mixed-gender groups [52]. However, the published article discusses results with respect to girls’ and boys’ groups only [52]. Nevertheless, it seems plausible that gender homophily structures children’s friendship networks in mixed-gender sports groups.

Although gender homophily is likely to be pronounced in children’s friendship networks in general, the relevance of certain similarities seems to be dependent on the setting. In organized sports, athletic ability plays a key role in practice and competition. This may lead to a clustering of athletes of similar ability levels, as indicated by a study on a professional women’s American football team [53]. In line with these findings, a longitudinal study in women’s collegiate basketball demonstrated a tendency for the starting players (usually considered the best players) to form a clique over the course of a season [54]. Homophily in athletic ability is likely to be a relevant factor in children’s friendship networks, too. However, such empirical studies specifically targeting children’s sports are missing.

In addition to the similarity in athletic ability, athletic ability has also proven to be relevant as an individual attribute in social networks. Regarding integrative organized sports with children with ID, the findings of Siperstein et al. [32] indicated a significant and positive correlation between hang-out-with nominations and athletic ability in individual sports (swimming) and in team sports (football) for both children with and without ID. They stated that in their program, the key for successful social relationships was athletic ability, not disability status [32]. However, their four-week summer program showed no correlation between new-friend nominations and athletic ability [32]. A possible explanation is that the length of the program was not sufficient for developing friendships. In the school setting, research has shown that higher athletic ability is correlated with being more popular [55,56,57].

Furthermore, language and migratory background structure friendship networks, as indicated by social network analyses in school and on sports [48,58]. Language is often considered in studies about the integration of people with migratory backgrounds or international students, and found to be critical for successful social participation in different life domains [59,60,61]. On the one hand, basic language skills in the national language (the language of the immigration country) are a condition for social participation [62]. On the other hand, language proficiency in the national language is seen as a facilitator for successful social interactions and friendships [60,61,62,63]. In a recent large-scale study, Cavicchiolo et al. [63] showed that proficiency in the national language is a predictor of social participation of first-generation and second-generation migrant children in school. The role of language in friendship networks in organized sports remains unexplored. However, how language proficiency affects sports and social participation is discussed. Here, inconsistent results prevail [64,65]. Further, findings in organized sports indicated that first-generation migrants experience fewer positive social relationships than second-generation migrants and natives [66,67,68,69]. Therefore, and considering the existing empiric research, language proficiency is likely to be a relevant factor for friendship networks in integrative organized sports and at school.

### 1.3. Goals of the Present Study

Even though research on organized sports, particularly integrative organized sports, is scarce, it is apparent that multiple factors play a role in friendship network formation. Thus, the first goal of the present study was to identify relevant individual attributes and dyadic factors in friendship networks in integrative organized sports with a special interest in ID. As the present study centered on antecedents of friendship networks as independent factors, the networks were treated as the dependent variable [70,71]. This network perspective allows for not only relating friendships to individual attributes but also to dyadic factors, while controlling for network structures [36]. Factors may play different roles in friendship networks in organized sports and school, as these two settings differ in several aspects: school is compulsory, school classes are given, children see each other five days a week for several hours a day and the performance in school has consequences for life after school, i.e., the main purpose of school is to acquire academic skills and be prepared for what comes afterward. Organized sports, however, are voluntary, children can change groups or even type of sports, they usually meet once/twice/three times a week for a few hours, and the main purpose of organized sports is to have fun while being physically active. Thus, the second goal was to determine whether friendship networks are formed differently in the two settings.

In this study, we addressed the following research questions:(1)How are friendship networks in integrative organized sports determined by exogenous factors, i.e., individual attributes (intellectual disability, athletic ability and language) and dyadic factors (gender homophily and similarity in athletic ability), while controlling for endogenous factors (network structures)? What role does the intellectual disability factor play, in particular?(2)Do differences exist regarding relevant factors in friendship networks in the two settings: integrative organized sports and integrative school? Does intellectual disability play a different role in friendship networks in integrative organized sports compared with integrative school?


Due to the deficient research on friendship networks in integrative organized sports, no hypotheses were formulated. Consequently, this study is exploratory in character.

## 2. Materials and Methods

### 2.1. Study Sample and Data Collection

This study is embedded in the Swiss National Science Foundation research project Social Participation in Sports (SoPariS) (2018–2021). Social participation is examined in two settings: school [15] and organized sports. For the research project, 109 integrative school classes (grades 3–6) and 31 integrative sports groups for organized sports were surveyed cross-sectionally across the German-speaking part of Switzerland. In school, children with ID (in the Swiss educational system, children with a Special Educational Needs (SEN) status because of intellectual disability are entitled to additional support in school; to evaluate a possible SEN status, a standardized assignment procedure is mandatory, with limited intellectual (IQ < 70) and adaptive functioning leading to a SEN status because of intellectual disability) were asked in a field questionnaire whether they participated in organized sports. When children with ID reported participating in integrative organized sports, their sports groups were visited with the consent of parents and coaches. All 32 children with ID in the 31 sports groups visited (one group had two children with ID) were also part of the school sample. For the present study, of the 109 classes, only the classes of the 32 children with ID visited in organized sports were considered. Thus, selection bias as a result of comparing different children with ID in the two settings was prevented. No sports group in the organized sports sample was part of a specific integrative or inclusive agenda or program. Due to incomplete answers and the method of choice for network analysis, the final study sample included a total of 24 sports groups and 24 school classes. Thus, seven groups and school classes were excluded. In five cases, the random graph models did not converge because the sports groups were too small (<10 children). Comparison of two more sports groups and classes was not possible because of missing answers in school questionnaires.

The description of the school sample revealed that gender was distributed evenly for children without ID, but not for children with ID (Table 1). As sports are often practiced in gender-homogeneous groups, the high number of boys with ID recruited through the school sample led to an uneven distribution of gender in the organized sports sample. Furthermore, it was apparent that children with ID mentioned Swiss-German or German less often as their main language spoken at home.

All children with and without ID completed a written questionnaire, instructed by at least two members of the research team. The children with ID received additional support either from a trained research assistant (organized sports/school) or special education teacher (school). Coaches and teachers were asked to complete an online survey.

### 2.2. Measures

Friendships were determined using quasi-unlimited peer nomination (nominations capped at nine, see [72]). Peer nomination is a common method to assess friendship networks [73]. Participants were asked to write down the names of their friends. This method seems to be more selective than to check name boxes [73] and it avoids negative selection [74]. Received and sent friendship nominations are called indegrees and outdegrees, respectively. Demographic information was obtained from the children’s questionnaire. They indicated their gender and main language spoken at home. The language was then coded in binary (0 = Swiss-German/German; 1 = other language). Information about ID was obtained beforehand through regional authorities and school principals. In addition, teachers and parents verified whether the respective child had a SEN because of their ID. Teachers and coaches were asked about the athletic ability of each participant compared with others of the same age (0 = below average, 1 = average, 2 = above average). Two dyadic factors were considered. Gender homophily was indicated by whether or not two individuals had the same gender. Similarity in athletic ability was measured as the absolute distance between individual values for athletic ability.

### 2.3. Data Analysis

The main interest of the present network analysis was the relevance of exogenous factors. However, when investigating networks, endogenous factors must also be considered [36]. Therefore, data were analyzed by means of exponential random graph modeling (ERGM) (R-package *ergm* [75]). ERGM is a method that compares an observed network with simulated graphs. It has some analogies to logistic regression as it predicts a binary outcome (friendship nomination present or not) [76]. Robins and Lusher [77] suggest to control for the following endogenous factors: density, reciprocity, popularity, activity, transitivity and two-path (Table 2).

To compare the social networks of integrative sports groups and school classes, in both settings, the same procedure was applied. The procedure for model selection was as follows: for each of the 48 friendship networks (24 school classes and 24 sports groups), a full model with all previously mentioned model parameters (for exogenous factors, see Section 2.2; for endogenous factors, see Table 2) was estimated. If the full model did not converge, the model parameters of popularity, activity, transitivity and two-path were removed and re-entered by a backward-forward model selection procedure [44]. Meta-analyses with the resulting 24 network models for each setting were then conducted [78] (using the R-package *metafor* [79]). Because of the backward-forward model selection procedure, not every model parameter could be included for every network model (see Table A1 in Appendix A). The same analyses as for these full models were then performed separately for individual attributes to check for their separate contributions. As in the school context, gender homophily is crucial for social networks; individual attributes were always estimated with all endogenous factors possible plus gender homophily. For a better interpretation of effect sizes, odds ratios were calculated by applying the exponential function to the regression coefficients for all exogenous variables in the full models [80].

Mixed-effects models were calculated to test whether intellectual disability plays a different role in friendship networks in organized sports compared with school (using the R-package *metafor* [79]). Therefore, the setting (organized sports vs. school) was used as a moderator variable on the incoming and outgoing ERGM-coefficients for intellectual disability. Because the ERGM-coefficients are a measure of the differences in friendship nominations between children with and without ID, it is analyzed whether these differences differ between the two settings.

## 3. Results

### 3.1. Descriptive Results

Figure 1 and Figure 2 offer a visual overview on the calculated friendship networks in organized sports and school. Sports groups were, on average, smaller than school classes. In organized sports, the smallest group was 10 and the largest group totaled 21 members. School classes varied from 12 to 24 children. In the mean, school classes were larger by four to five children than sports groups (Table 3). The descriptive statistics for gender, language and intellectual disability are reported in the sample description (Table 1); Table 3 provides descriptive results for further variables included in ERGMs.

The descriptive results indicated that children without ID scored higher in athletic ability than children with ID in both settings. Indegree values in organized sports showed that children with ID received nearly as many friendship nominations as their peers without ID, whereas in school, children with ID received fewer nominations than their peers.

Regarding endogenous factors, the visualized networks (Figure 1 and Figure 2) revealed apparent differences in the variance in the density between the two contexts. In sports groups, the variance was considerably higher than in school classes (Figure A1 in Appendix A). However, comparing the mean values for the density and reciprocity of networks of sports groups and school classes, only slight differences were observed (Table 3). The variance in reciprocity is similar between the two contexts, but within each context, large differences in reciprocity can be observed. In organized sports, the lowest reciprocity is 0.45 and the highest 0.79, i.e., in one network, only 45% of nominations are reciprocated, whereas in the other, 79% are reciprocated.

### 3.2. Exogenous Factors in Friendship Networks in Integrative Organized Sports

Table 4 contains the random-effects models for friendship networks in the integrative organized sports sample. All control variables, i.e., endogenous variables, showed significant effects in all organized sports models. The significant effects of the endogenous variables indicated that the estimated network structures played a significant role in the friendship networks, underlining the importance of controlling for these endogenous factors. For instance, the significant positive effect of reciprocity indicates that teammate *i* nominates teammate *j* more frequently than by random chance, given that *j* nominates *i*.

Regarding individual attributes, effects for both indegree and outdegree were estimated. Language seemed not to be a relevant exogenous factor (OSM 2 and 5), whereas the effects of athletic ability were significant and positive (OSM 3 and 5). This means that children with a higher value in athletic ability received (indegree) and sent (outdegree) more friendship nominations than children with a lower value. The odds ratios for the full model (OR OSM 5) indicated that a child scoring higher by one in athletic ability was 1.22 times more likely to be nominated as a friend than a child scoring one lower (indegree), and it was 1.13 times more likely that a more athletic child would send a friendship nomination (outdegree). Regarding dyadic exogenous factors, similarity in athletic ability was not a determinant of friendship networks in organized sports (OSM 3 and 5). The weak significant effect of gender homophily in the full model (OSM 5) indicated its importance in friendship networks in gender-mixed integrative sports groups. Same-gender friendships were 1.3 times more likely to occur than opposite-gender friendships, as indicated by the odds ratio for gender homophily. This means that friendships in organized sports groups were formed around the criterion of same gender, but an equal or similar athletic ability was not a constituting criterion for friendship formation. However, the results regarding gender homophily must be interpreted carefully, as only 6 of the 24 sports groups were mixed-gender groups.

The effects regarding intellectual disability were of particular interest in the present study. The non-significances of the effects of intellectual disability indicated that when controlling for all endogenous factors and gender homophily (OSM 4), and for all endogenous factors and all other exogenous factors (OSM 5), intellectual disability was not relevant in friendship networks in integrative organized sports. The positive values indicated that intellectual disability had a positive but insignificant effect on friendship formation.

### 3.3. Exogenous Factors in Friendship Networks in Integrative School and Comparison with Integrative Organized Sports

Table 5 presents random effects models for friendship networks in the integrative school setting. As in integrative organized sports, in all school models, endogenous factors were found to have significant effects, indicating that they were also important to control for in the school setting.

With respect to dyadic factors, positive and significant effects for gender homophily were observed in all school models. The odds ratio for gender homophily in the full model (OR SCM 5) indicates that gender-homogenous friendships were 6.05 times more likely than friendships with the opposite gender. The much higher odds ratio compared with organized sports indicates that in school, the phenomenon of girls more often befriending other girls, and boys other boys, was considerably more pronounced in school classes than in gender-mixed sports groups. Similarity in athletic ability was not a relevant factor in either setting, although in school, this coefficient almost reached a weak significance (*p*-value of 0.106), while in organized sports, this was not the case (*p*-value of 0.674). Unlike in sports groups, in school classes, children who spoke Swiss-German or German at home received more friendship nominations than the others, as indicated by the negative and significant effect for language (indegree). Findings for athletic ability as an individual attribute on received friendships (indegree) were similar to the results in organized sports. However, in school, unlike in organized sports, athletic ability had no significant effect for sent nominations (outdegree).

In the school sample, significant and negative effects were revealed for intellectual disability. When tested separately (SCM 4), intellectual disability was a significant predictor of lower incoming friendship nominations. This effect persisted when controlling for all effects at once (SCM 5). Thus, findings for intellectual disability in integrative school differed from the findings in integrative organized sports.

### 3.4. Moderator Analyses

By means of mixed-effects analyses, it was tested whether the ERGM-coefficients for intellectual disability differ significantly between the two settings. Table 6 presents the estimated mixed-effects coefficients. The results indicated that although children with ID sent slightly more friendship nominations than children without ID in organized sports (0.09), and sent less in school (−0.11), this difference was not significant (*p*-value of 0.37). On the other hand, the difference differed significantly between the two settings for received friendship nominations (*p*-value of 0.05).

## 4. Discussion

The main interest in this study was identifying relevant factors in friendship networks in organized sports. As positive relationships in social groups are crucial for psychosocial health, and as people with ID are found to be on the verge of society in different life domains [8,9,81], we emphasized the role of intellectual disability. Additionally, a comparison between organized sports and school allowed us to gain insight into the importance of various factors in the two settings. As opposed to most previous studies in and outside organized sports, we applied a genuine network perspective to friendships. This innovative focus enabled us to consider three individual attributes and two dyadic factors while simultaneously controlling for the effects of network structures.

Of all factors examined, intellectual disability was found to be an irrelevant factor in friendship networks in organized sports, as there were no significant effects either when tested separately or when controlling for all other exogenous factors. These quantitative results add weight to existing research [28,29,30,31,32,81], indicating that friendships beyond ID are formed in integrative organized sports. A possible reason for why children with ID are a part of friendship networks in a manner equal to their peers is that organized sports provide joint activities. Physical proximity is thought of as the most basic source of homophily, enabling the discovery of similarities and shared interests, and therefore, influencing friendship networks [49]. Maybe practicing sports together is “enough of a similarity,” as “sharing the same passion for a sport” is not restricted by (dis)ability [82,83].

The results further indicate that athletic ability is an important exogenous factor in friendship networks in integrative organized sports. The relevance of athletic ability persisted when controlling for all other included factors. Therefore, and in line with Siperstein et al.’s [32] findings for hang-out-with nominations, athletic ability is important for social relationships. However, whereas Siperstein et al. [32] found no correlation between athletic ability and new friendship nominations in their four-week program, the results of the present study indicate that for friendship networks in year-round organized sports, athletic ability is important. The high popularity of better athletes in a group could be due to their importance for the groups’ sporting success, especially in team sports. Furthermore, previous research showed that athletic children, in particular, assume leadership roles in groups [84].

More athletic children were also found to be more active in friendship networks, meaning they sent more friendship nominations. Even though more athletic children in a group are more popular and more active, this does not associate with friendship networks being divided into more athletic children on one side and less athletic children on the other side, as similarity in athletic ability was not a significant factor. A possible explanation for this finding is that the common interest in the type of sport and the common pursuit of sporting goals can build bridges beyond level of ability. In addition, and especially in competitive club sports, the clear affiliation with a sports club (“same shirt”) and the competitive nature toward other sports clubs and groups can lead to a more positive evaluation of all in-group members compared with members of other groups, which is a central assumption of social identity theory [85], thus uniting children of different ability levels in the same group.

In gender-mixed sports groups, friendship networks are also structured by gender, as indicated by the significant effects of gender homophily in the full model. The finding that gender structures friendship networks aligns with previous research in education [48,49,50,51] and provides first insights into gender homophily in organized sports. Yet, it is important to note that when tested for gender homophily separately, the effect was non-significant. Furthermore, caution is advised regarding the interpretation, as only one-quarter of the 24 sports groups were mixed-gender groups.

The comparison between integrative organized sports and integrative school provides further insight into the integrative potential of organized sports. Commonalities as well as differences between the settings regarding relevant factors for friendship networks were revealed. Athletic ability, gender homophily, language and intellectual disability were all significant factors in friendship networks in school. Thus, the results are in line with the existing body of literature [12,14,55,57,60]. Of those four, the importance of athletic ability and gender homophily prevailed in both settings, underlining their importance in friendship networks. While high athletic ability in sports groups might be a favorable attribute because of its contribution to sporting success, in non-sporting contexts, it might just be a favorable attribute that children like in other children [55].

Differences in the findings in organized sports and school were evident for intellectual disability. The negative effects of intellectual disability in friendship networks in school suggest that intellectual disability plays a different role in school than in organized sports, as indicated by the meta-analyses. The moderator analyses supported this finding for received friendship nominations, but not for sent nominations. Relative to children without ID, children with ID received more friendship nominations in organized sports than in school. As the school sample covered the same children with ID as the organized sports sample, the difference in effects cannot be explained through differences in the ID of the children. An explanation for the difference in effects could be that intellectual disability is more relevant and more visible in school. In school, children with intellectual disabilities receive additional support from SEN teachers. Additionally, due to their limited cognitive abilities, they receive different learning objectives than their classmates without ID. Therefore, differences between children with and without ID are more apparent in school than in organized sports. Additional meta-analyses were conducted with academic ability as an independent variable instead of athletic ability (not shown here). These further analyses showed that academic ability as an individual attribute is not a relevant factor for friendship formation in school, but is as a dyadic factor, i.e., similarity in academic ability structures friendship networks (in line with [86,87,88]). The effect of intellectual disability was still significantly negative, but less prominent when academic ability was included in the model due to their content-related overlap.

In the present study, differences in effects were found not only for intellectual disability but also for the main language spoken at home. In school, children speaking German or Swiss-German at home were more popular than children speaking other languages, whereas language did not matter in organized sports. These differences might be related to the different functions of language skills in the two settings. Even though in organized sports, basic language skills are necessary for communication, they are less relevant for sporting success, i.e., children with poor language skills can also contribute positively to the sporting outcome of a team. In contrast, language proficiency is important in school. For example, children with inferior language skills might be seen as a barrier to success in group tasks, which might also influence friendship relations.

Seemingly, intellectual disability and language are less important in organized sports than in school. Thus, the findings nourish the hope that organized sports can act as a social glue, holding together different social groups in an increasingly diverse society. Furthermore, post hoc analyses of the complete school sample (109 classes) revealed that regardless of their participation in organized sports, children with ID are at risk of social exclusion in the integrative school setting. Interestingly, children with ID that participated in organized sports were generally worse off regarding friendship nominations in school than the children with ID that did not participate in organized sports. Consequently, particularly for children with ID who struggle to bond in integrative school, organized sports can be a valuable place to make friends.

The differences in the factors relevant in friendship networks in organized sports and school might be attributable to the societal functions of the sporting system and the educational system. Functions assigned to school are, amongst others, qualification, selection, legitimation and socialization [89,90]. School contributes largely to individual development, but might create an otherness of children with ID, making disability more visible and tangible. Discussions on ableist assumptions in education view this as harmful for the social participation of children with ID and other disabilities (e.g., [91]). The relevance of the sporting system is also defined by its capability to solve important societal challenges [92]. Societal functions of recreational sports are: socialization, integration and health promotion [93,94] (for club sports [18]). Considering people with disabilities and sports, political debates have emerged on the contribution of organized sports toward fostering their health and social participation. To facilitate sporting activities where people with disabilities participate as equal members, national and international conventions, most prominently the CRPD, have been adopted. Nowadays, providing sporting activities for people with disabilities is also a responsibility of mainstream sports organizations [95].

Over the years, different types of structural integrations of people with disabilities in organized sports have been identified [96]. In contrast to previous research, the sports groups surveyed in the present study were neither part of a specific time-limited program with the goal of fostering contact between children with and without ID (e.g., summer camp [32]) nor part of sports activities provided or supported by Special Olympics (e.g., [29,33,81,97]). The present findings stem from regular year-round sports activities in local organizations, also referred to as mainstream sports. The organizations providing the activities did not follow a specific inclusive agenda, as indicated by surveyed coaches. Therefore, the organized sports sample in the present study can best be assigned to “direct integration” [96] because children with and without disabilities practice sports side by side and as equal individuals. Consequently, the findings for intellectual disability support the integrative potential of organized sports in the “direct integration” setting. No conclusions can be drawn on other types of integrative organized sports, such as “organizational integration” [96], where people with (intellectual) disabilities practice sports in separated sports groups, for example, in “special teams”.

The present study has limitations. A selection bias could have affected the results. In general, people with disabilities are found to be underrepresented in organized sports [35,98,99,100,101]. For people with [102] and without disabilities [103,104], building friendships is an important motive for entering and remaining in organized sports. If expectations based on this motive are not met, meaning that insufficient friendships are formed or negative experiences (e.g., bullying) prevail, withdrawal from the sports group may occur. This is not only the case for children with ID, but as they are more often confronted with stigmas and prejudices, it might be more often the case for them. The literature suggests that intrapersonal and interpersonal constraints are associated with dropout from organized sports among all children [105]. Regarding children with ID only, a study on Special Olympics athletes identified that not only changes in interest but also limited program availability as causing dropouts amounting to over one-third of the athletes [106]. Children with ID who dropped out of sports groups could not be included in the study. Also, the voluntary nature of organized sports and the compulsory nature of school may have affected the results of the study. When experiencing social exclusion, the bar for leaving the group is lower in voluntary settings. Therefore, studies investigating withdrawal of children with ID in (non-program-based) integrative organized sports are needed.

As the organized sports sample had significantly more boys than girls, and only three all-girl sports groups participated in the study compared with nine all-boy groups, the question arises as to how this imbalance affects our conclusions for girls. Regarding athletic ability, previous studies in school demonstrated that this factor seems to be more relevant for friendships between boys than for girls [107]. However, the results for the school sample in the present study indicated that athletic ability was also a significant factor in school, where gender is distributed evenly. In addition, post hoc analysis for the three all-girl sports groups showed inconsistencies regarding the importance of athletic ability, and the results for similarity in athletic ability are in line with the findings for the study sample. Nevertheless, the role of athletic ability in friendship networks in all-girl sports groups has yet to be examined. Moreover, studies investigating friendship networks in mixed-gender sports groups are needed.

In the study, language was assessed by inquiring the children’s main language spoken at home. We used this variable as a proxy for language proficiency, arguing that it affects social participation, as indicated by literature. Here, a language test would more accurately capture language proficiency.

Keeping in mind the heterogeneity of ID, the present findings apply primarily to children with mild ID. Only in a few cases was close individual guidance in sports needed, as indicated by coaches. Some coaches did not even know about the ID of the child in the sports group. The children with IDs all attended regular primary schools and were, therefore, used to interacting in a group with children without ID. Furthermore, we explored friendship networks on the basis of quantitative data. Consequently, conclusions about the quality of friendships were not possible. Future studies may provide insight into different types of social support provided by friendships, i.e., emotional, instrumental, informational and appraisal support [3].

## 5. Conclusions

In conclusion, children with ID are a part of friendship networks in sports groups in a manner equal to their peers without ID. Therefore, in organized sports, children with ID can establish and participate in positive social relationships, i.e., friendship networks, which foster social support and psychosocial health [1,2]. Linking the revealed importance of athletic ability and the non-significance of intellectual disability, it seems to be beneficial for the social participation of children with ID that they find sporting opportunities where they are among the better athletes of the group. The voluntary nature of organized sports allows for choosing a sports activity purposefully, so it matches individual inclinations of the child with ID, and so, at least to a certain degree, one can estimate in advance that their athletic ability is on par with that of other children in the group. Examples of exclusion experiences of people with ID—e.g., “the swim coach will not let me train with the squad as I am too slow; he makes me swim in the junior class where they are all younger than me and do not do racing training, which is what I want” [108] (p. 399)—emphasize the importance of a fitting of athletic ability for successful participation. In addition, being part of a friendship network in sports may lead to lasting positive physical health outcomes because friends practicing sports may maintain or enhance their willingness to be physically active [2]. To facilitate benefits to psychosocial and physical health, children with ID should be encouraged by policy makers, practitioners and parents to participate in integrative organized sports.

## Figures and Tables

**Figure 1 ijerph-18-06603-f001:**
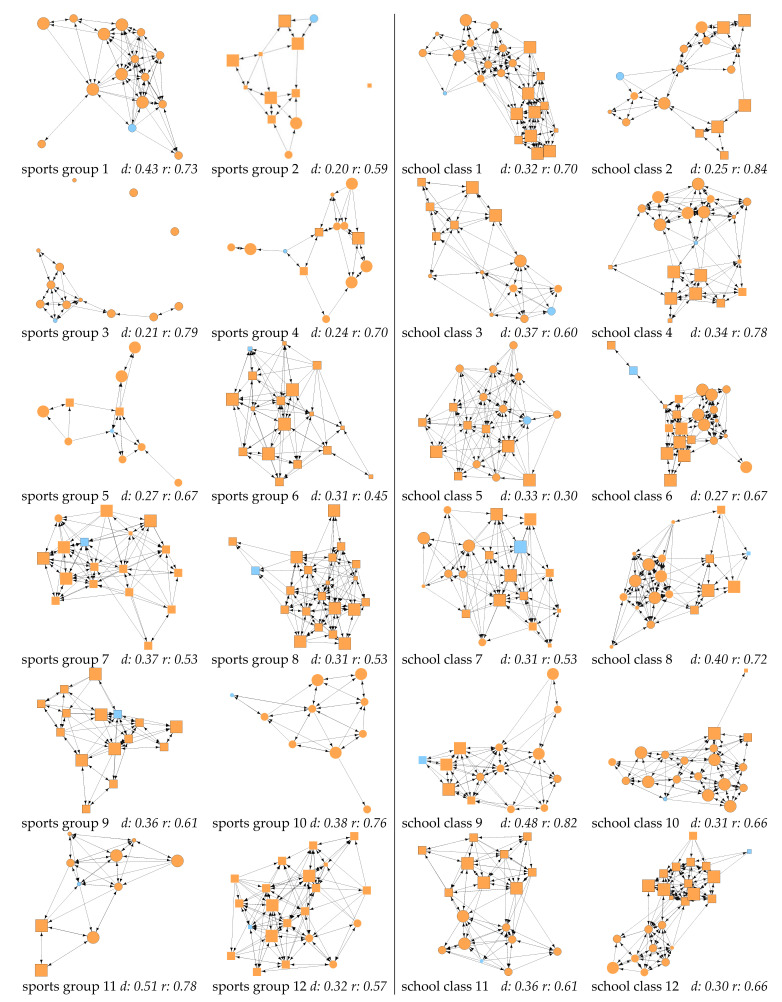
Friendship networks in integrative organized sports and school. Square = boy; circle = girl; blue = child with ID; orange = child without ID; node size = athletic ability; black border = speaks Swiss-German or German at home; no border = other languages; ties = friendship nominations.

**Figure 2 ijerph-18-06603-f002:**
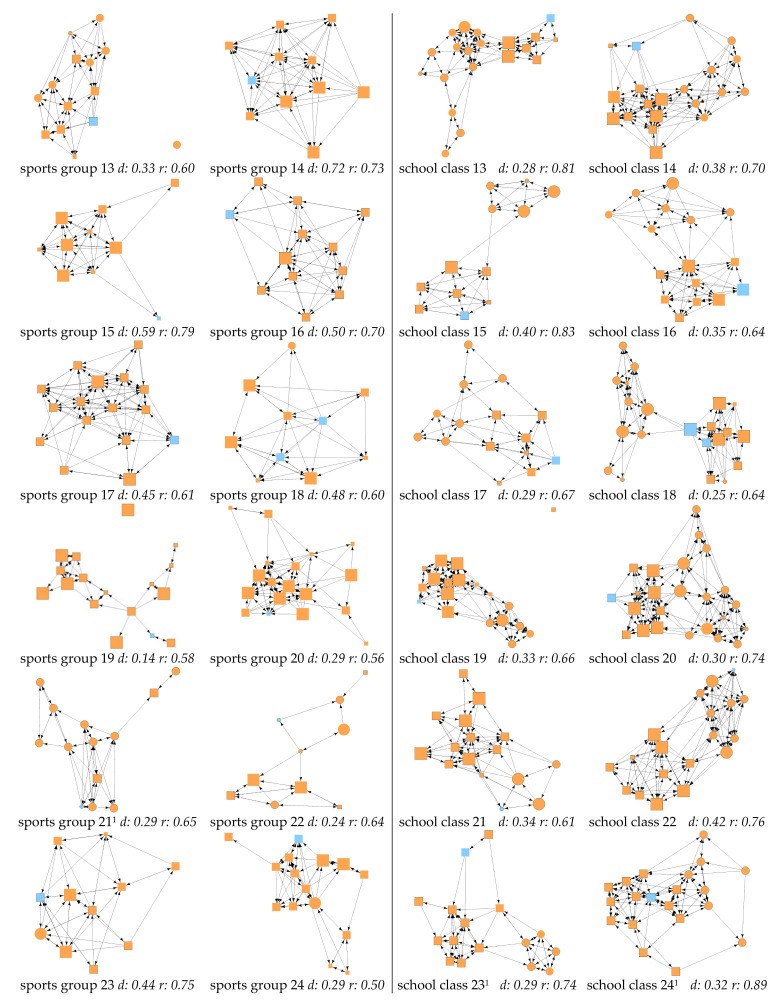
Continued friendship networks. ^1^ With sports group 21, school class 23 and school class 24, missing values in athletic ability were replaced with the average for visualization purposes. For model estimation, athletic ability was not included with the aforementioned networks (see Table A1 in Appendix A).

**Table 1 ijerph-18-06603-t001:** Description of sample (N = 24 sports groups, N = 24 school classes).

	Organized Sports		School	
	Children without ID ^1^	Children with ID ^1^	Children without ID ^1^	Children with ID ^1^
n	306	25	416	25
Age (M, SD)	11.31 (2.24)	12.27 (1.31) ^3^	11.31 (1.06)	11.85 (0.95)
Gender (% girls)	88 (28.8%)	9 (36%)	209 (50.2%)	9 (36%)
Language (% CHger/GER ^2^)	256 (83.7%)	16 (64%)	334 (80.3%)	16 (64%)

^1^ Intellectual disability. ^2^ Percentage of children speaking Swiss-German (CHger) or German (GER) at home. ^3^ The sports groups were visited, on average, five months later than the school classes.

**Table 2 ijerph-18-06603-t002:** Endogenous factors.

Model Parameter	Description	Visualization
Density	Density indicates the general probability of the presence of a friendship nomination in a network. A coefficient of zero means that exactly half of all possible nominations are sent; that density is 0.5. The density coefficient is comparable to the intercept in a regression analysis.	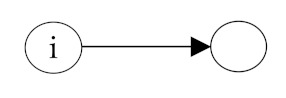
Reciprocity	Reciprocity indicates the probability that a sent friendship nomination is reciprocated. A positive coefficient means that the probability of a nomination from child *j* to *i* increases given that a nomination from child *i* to *j* exists.	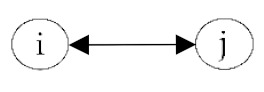
Popularity	Popularity is a measure of indegrees. A positive coefficient indicates that there are children in the network that are more popular than the others, i.e., some children receive more friendship nominations.	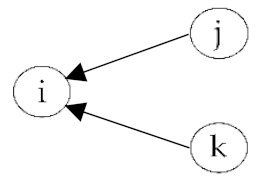
Activity	Activity is a measure of outdegrees. A positive coefficient indicates that there are children in the network that are more active than the others, i.e., some children send more friendship nominations.	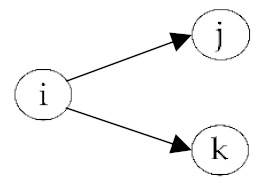
Transitivity	Transitivity indicates the probability that we observe the structure “a friend’s friend is also my friend”, i.e., the probability of an existing friendship nomination from child *i* to *k* given that child *k* is a friend of *j* and child *j* a friend of *i*.	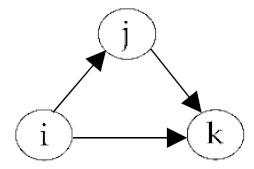
Two-Path	Two-path is used as a control structure to correctly estimate the effect of transitivity.	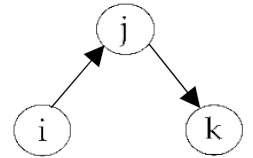

**Table 3 ijerph-18-06603-t003:** Descriptive results (N = 24 sports groups, N = 24 school classes).

	Organized Sports		School	
	Children without ID ^1^	Children with ID ^1^	Children without ID ^1^	Children with ID ^1^
n	306	25	416	25
Athletic ability 0–2 (M, SD)	1.16 (0.64)	0.52 (0.51)	1.20 (0.63)	0.75 (0.68)
Indegrees (M, SD)	4.65 (2.78)	4.44 (2.81)	5.95 (2.68)	3.32 (2.58)
Outdegrees (M, SD)	4.66 (2.69)	4.36 (2.5)	5.85 (2.21)	5.00 (2.27)
Density (M, SD)	0.36 (0.13)		0.33 (0.06)	
Reciprocity (M, SD)	0.64 (0.10)		0.69 (0.12)	
Group size	13.79 (3.44)		18.38 (3.06)	

^1^ Intellectual disability.

**Table 4 ijerph-18-06603-t004:** Random-effects models for friendship networks in organized sports.

Effect	OSM 1	OSM 2	OSM 3	OSM 4	(Full) OSM 5	OR OSM 5
Density	−2.59 ***	−2.50 ***	−2.27 ***	−2.53 ***	−2.57 ***	
Reciprocity	1.87 ***	1.87 ***	1.81 ***	1.96 ***	1.84 ***	
Popularity	0.19 ***	0.15 ***	0.17 ***	0.17 ***	0.13 ***	
Activity	0.17 ***	0.15 ***	0.16 ***	0.17 ***	0.14 ***	
Transitivity	0.70 ***	0.76 ***	0.67 ***	0.72 ***	0.68 ***	
Two-Path	−0.14 ***	−0.14 ***	−0.16 ***	−0.16 ***	−0.16 ***	
Gender Homophily	0.21 (0.14)	0.27 (0.2)	0.26 (0.15) ^+^	0.19 (0.14)	0.26 (0.15) ^+^	1.30
Sim. Athletic Ability			−0.01 (0.06)		−0.03 (0.07)	0.97
Language (indegree)		−0.01 (0.13)			−0.10 (0.18)	0.91
Language (outdegree)		0.01 (0.14)			−0.01 (0.15)	0.99
Athletic Ability (in.)			0.15 (0.05) **		0.20 (0.06) **	1.22
Athletic Ability (out.)			0.07 (0.06)		0.12 (0.06) *	1.13
Int. Disability (in.)				0.05 (0.14)	0.18 (0.19)	1.20
Int. Disability (out.)				0.04 (0.13)	0.09 (0.16)	1.09

*p* values: *** 0.1%, ** 1%, * 5%, ^+^ 10%; OSM = organized sports model; OR = odds ratio.

**Table 5 ijerph-18-06603-t005:** Random-effects models for friendship networks in school.

Effect	SCM 1	SCM 2	SCM 3	SCM 4	(Full) SCM 5	OR SCM 5
Density	−3.78 ***	−3.84 ***	−4.09 ***	−3.66 ***	−3.80 ***	
Reciprocity	2.03 ***	2.01 ***	1.89 ***	2.04 ***	2.03 ***	
Popularity	0.22 ***	0.21 ***	0.1 ***	0.20 ***	0.17 ***	
Activity	0.09 ***	0.08 **	0.09 **	0.08 **	0.06 *	
Transitivity	0.62 ***	0.63 ***	0.60 ***	0.60 ***	0.53 ***	
Two-Path	−0.10 ***	−0.09 ***	−0.11 ***	−0.10 ***	−0.10 ***	
Gender Homophily	1.66 (0.14) **	1.83 (0.16) ***	1.78 (0.16) ***	1.70 (0.14) ***	1.80 (0.15) ***	6.05
Sim. Athletic Ability			−0.09 (0.04) *		−0.08 (0.05)	0.93
Language (indegree)		−0.29 (0.1) **			−0.21 (0.09) *	0.81
Language (outdegree)		−0.02 (0.1)			0.03 (0.11)	1.03
Athletic Ability (in.)			0.26 (0.05) ***		0.23 (0.05) ***	1.26
Athletic Ability (out.)			0.10 (0.06) ^+^		0.11 (0.07)	1.11
Int. Disability. (in.)				−0.48 (0.17) **	−0.41 (0.21) *	0.66
Int. Disability (out.)				−0.03 (0.14)	−0.11 (0.15)	0.90

*p* values: *** 0.1%, ** 1%, * 5%, ^+^ 10%; SCM = school model; OR = odds ratio.

**Table 6 ijerph-18-06603-t006:** Mixed-effects meta-analyses for intellectual disability with the setting as the moderator variable.

Effect	Organized Sports	School	*p*-Value
Int. Disability (in.)	0.17	−0.39	0.05
Int. Disability (out.)	0.09	−0.11	0.37

## Data Availability

The data presented in this article are not yet available due to an agreement with the Swiss National Science Foundation to publish the data in an open-access repository after the end of the project. Please direct requests on the availability of data to the corresponding author.

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
