# Peer review of "Friendships in Integrative Settings: Network Analyses in Organized Sports and a Comparison with School"

_ijerph, 2021, doi:10.3390/ijerph18126603_

Round 1

Reviewer 1 Report

after to read carrefully I think this manuscript is suitable for publish

Reviewer 2 Report

I want to thank the authors for their contribution in this field of study. 

Specific comments 

I think that all parts of the current manuscript are well written and support the aim of the manuscript. 

I think that the current manuscript can publish as it is. 

Reviewer 3 Report

Friendships in Integrative Settings

I find this a relevant and timely study, well-written and well researched.  I think this should be a relevant paper for IJERPH that could be published with minor revisions.

How sports work for disabled people is a much studied and discussed topic these days, and this study provides an important and useful study of how ID athletes are socially integrated through their participation in organized sports. The literature review seems to cover relevant research.

The study is based on a solid dataset (from school and sports).

As an improvement compared to previous research, network analyses are used so we learn how a set of exogenous factors (ID, gender, language) matter for establishment of social networks controlled for endogenous and dyadic factors.

The theoretical foundation of the paper could perhaps have been more explicit and systematic and better integrated in the analyses?  

The empirical analyses as competently conducted.

It is discussed in the limitation part, but one could wonder how important the differences between a compulsory school and volunteer sports field are and how the leave-mechanisms matter for the result (if you do not succeed socially in sports you leave, if you do not succeed in school socially, you stay on).

But, all in all, a good article.

Reviewer 4 Report

- The topic is interesting and both the research question and the empirical form of analysis can be described as innovative and purposeful.
- Define right at the beginning how ID is defined here
- Clarify the structure in the theory section (insert subheadings).
- Explain some sentences better or substantiate claims better. 
- The first statements about SNA on p. 2 are too presuppositional, too technical, and should be supported with a substantive example.
- Some sentences are poorly formulated and therefore incomprehensible (e.g., To specify the exogenous factors that are causally related to friendship networks, an informed decision is required [40], p. 2)
- Regarding the exogenous factors and the question of possible causal mechanisms of action should be supported by a theoretical framework model - so far only different empirical findings are listed here.
- The comments on the factor "language" (p. 2) are not convincing, to what extent would this be practically relevant here? And wouldn't the quality of linguistic expression be more interesting instead of language per se?
- Although the differences between school and organized sports settings are clearly presented, it is not yet clear which factors should be more relevant in one setting and which in the other - more theoretical explanations should be added.
- Sample, data collection and measures are adequately described
- The description of the analysis procedure in 2.3 is too technical. Readers who are not familiar with the method can only follow superficially - therefore please supplement with content-related examples in order to be able to better assess the relevance of the individual parameters.
- Do I understand it correctly that always only 1 ID child is represented in the respective groups? Why is this the case?
- How exactly the statistical procedure was to arrive at the ORs in table 4 is not clear to me - perhaps the authors could explain this a bit more precisely and comprehensibly
- The network illustrations of the individual groups are exciting and helpful and certainly a strength of the manuscript.
- The method is very sophisticated already, but I wondered if the comparison sport-school could not also be tested as a moderator variable

Round 2

Reviewer 4 Report

The authors have thoroughly revised their manuscript. I am very convinced by all these changes and think the paper has been much improved through revision. In the present form, I believe it can make a valuable contribution to the literature.